# Microstructural Characteristics of High-Pressure Die Casting with High Strength–Ductility Synergy Properties: A Review

**DOI:** 10.3390/ma16051954

**Published:** 2023-02-27

**Authors:** Qiang Yang, Xiaohan Wu, Xin Qiu

**Affiliations:** 1State Key Laboratory of Rare Earth Resource Utilization, Changchun Institute of Applied Chemistry, Chinese Academy of Sciences, Changchun 130022, China; 2Key Laboratory of Superlight Material and Surface Technology, Ministry of Education, College of Materials Science and Chemical Engineering, Harbin Engineering University, Harbin 150001, China

**Keywords:** Mg alloys, high-pressure die casting, strength, ductility, intermetallic phase

## Abstract

In response to the increased emphasis on reducing carbon emissions, the demand for lightweight, high-performance structural materials is quickly increasing, and Mg alloys, because of their having the lowest density among the common engineering metals, have demonstrated considerable advantages and prospective applications in contemporary industry. High-pressure die casting (HPDC), owning to its high efficiency and low production cost, is the most extensively utilized technique in commercial Mg alloy applications. The high room-temperature strength–ductility of HPDC Mg alloys plays an important role in their safe use, particularly in the automotive and aerospace industries. With respect to HPDC Mg alloys, their mechanical properties highly rely on their microstructural characteristics, particularly the intermetallic phases, which are further dependent on the alloys’ chemical compositions. Therefore, the further alloying of traditional HPDC Mg alloys, such as Mg-Al, Mg-RE, and Mg-Zn-Al systems, is the most adopted method to further improve their mechanical properties. Different alloying elements lead to different intermetallic phases, morphologies, and crystal structures, which can have helpful or harmful effects on an alloy’s strength or ductility. The methods aimed at regulating and controlling the strength–ductility synergy of HPDC Mg alloys have to arise from an in-depth understanding of the relationship between the strength–ductility and the components of the intermetallic phases of various HPDC Mg alloys. This paper focuses on the microstructural characteristics, mainly the intermetallic phases (i.e., components and morphologies), of various HPDC Mg alloys with good strength–ductility synergy, aimed at providing insight into the design of high-performance HPDC Mg alloys.

## 1. Introduction

Due to the pressing demand for lightweight products in the automobile and aerospace industries, light structural materials with high performance are increasingly being developed in response to the increased focus on lowering carbon emissions [1]. It is well known that Mg alloys are the lightest among the common engineering metals, with a density that is only one-quarter that of steel and two-thirds that of aluminum [2]. Furthermore, they generally have a relatively high specific strength [3,4] and other special physical properties [5]. Therefore, Mg and its alloys offer significant potential use in many industrial marketplaces [6]. Generally, they may be readily cast to a near-net shape by conventional casting methods because of their relatively low melting temperature [7,8,9]. According to incomplete statistics, more than 90% of structural parts made of Mg alloy are prepared by casting methods [7,10]. Among these commercial methods, high-pressure die casting (HPDC) is the most widely used technique (over 85%) [7,8,11,12,13,14].

HPDC is a proven manufacturing process for Mg alloys by which products are produced in volume, generally as large automobile parts, such as car seats and car doors [7,10,15]. Usually, it is a repetitive process where identical products are cast at a high production rate by injecting molten metal under high pressure (MPa) into a steel die, ordinarily with an oil heating system. As a result, the melt rapidly cools until the solidified part is sufficiently rigid to permit ejection from the mold, with a short cycle time of tens of seconds. Moreover, HPDC can make parts to the final shape without extra machining or other operations, producing tighter and dimensionally stable Mg-based specifications. These factors endow HPDC technology with high efficiency and cheap production costs [7]. On the other hand, defects such as gas pores and shrinkage unavoidably form in castings [16]. Hence, a large amount of work has been carried out in order to reduce or even eliminate these die casting defects by controlling the melt and die temperature [17], optimizing the HPDC parameters [14,18,19], or using a vacuum-assisted system [20]. Generally, defects seriously deteriorate the mechanical properties of HPDC Mg alloys, and the differences in the properties reported by various authors might be from these defects. However, considering the main topic of intermetallic phases, the impact of the optimization of the HPDC parameters on the microstructural and mechanical properties has not previously been discussed.

Not all Mg alloys are suited for the HPDC method. HPDC Mg alloys generally require excellent castability, and the most common system is Mg-Al. The presence of Al can simultaneously increase an alloy’s strength and die-casting performance, leading to the majority of the commercial applications for AZ91 (Mg-9Al-1Zn, wt%) [21,22] and AM50/60 (Mg-5/6Al-0.2Mn, wt%) [23,24]. Nonetheless, the traditional HPDC AZ91 and AM50/60 alloys have moderate strength, particularly when the temperature is above 120 °C and/or there is poor ductility, which seriously limits their application [7,25]. Aimed at modifying the mechanical performance of the Mg-Al system, especially at high temperatures, further alloying has been widely adopted, such as using Si [26,27], Ca [28], and/or Sr [29,30,31]; RE [13,32,33,34,35]; much more Zn [36,37,38]; or two other elements [39,40,41,42]. Two classic series that are used in automobile industries were developed as Mg-Al-Ca/Sr (AX/J)-based alloys [43,44,45,46,47,48,49] and Mg-Al-RE (AE)-based alloys [50,51,52,53,54]. For example, the representative commercial AE44 (Mg-4Al-4RE-0.4Mn, wt%) alloy has excellent room temperature properties and creep resistance even at 200 °C, and it has been used as the engine cradle for the GM Corvette [14,15]. Furthermore, HPDC AX/J alloys have also been widely used in structural applications in automobile industries [14,15]. Moreover, Mg-RE-based and Mg-Zn-Al (ZA)-based HPDC alloys have excellent mechanical properties, also having great potential in commercial applications.

In engineering applications, the strength–ductility synergy of the structural materials plays an important role in their safe use [55,56,57]. However, the low strength and poor ductility of Mg alloys seriously limits their wide application, particularly in power train components [57]. Generally, excellent strength–ductility synergy has frequently been observed in wrought Mg alloys, particularly in Mg-RE [58,59,60], Mg-Al [55,57], Mg-Li [61,62], and Mg-Zn [62,63,64] systems. Nonetheless, progress in their preparation is not expected in many cases. Thus, developing HPDC Mg alloys with high strength and high ductility could dramatically promote the application of Mg alloys, especially in the automobile, electronic, and aerospace industries [14,65]. As is well known, HPDC technology generally results fine grains of 5–16 μm, and the mechanical properties of HPDC Mg alloys highly rely on their intermetallic characteristics, such as the crystal structures of the intermetallic phases [66], the intermetallic skeleton morphologies [67,68], and in some cases the solid solutes or precipitates [69]. This paper reviews the strength and ductility of various HPDC Mg alloys at room temperature and states the corresponding microstructural characteristics, aimed at serving as a guide for the future development of HPDC Mg alloys with high strength and high ductility.

## 2. Strength–Ductility Properties of HPDC Mg Alloys

Figure 1 summarizes the tensile yield strength (TYS) versus elongation to fracture of various HPDC Mg alloys at room temperature [7,40,42,46,48,50,54,65,66,68,69,70,71,72,73,74,75,76,77,78,79,80,81,82,83,84,85,86,87,88,89,90,91,92,93,94,95,96]. It is clear that the Mg-Al- and AE-based alloys have relatively better ductility than Mg-RE-, ZA-, and AX/J-based alloys. As is well known, AZ91 and AM60 alloys are the most representative commercial HPDC Mg alloys, and the AZ91D alloy generally has relatively high strength, while the AM60 alloy has excellent ductility [97,98]. Herein, we defined the strength–ductility level of AZ91 and AM60 alloys as the Mg-Al-system level, highlighted by an orange, dotted line in Figure 1. It is clear that the AE-based alloys generally had better strength–ductility synergy than the Mg-Al-system level, and the AE42 (Mg-4Al-2RE, wt%) alloy exhibited a satisfactory strength–ductility level with relatively good ductility. Meanwhile, the other HPDC alloys, such as the Mg-RE series, AX/J series, and ZA series have outstanding high strength. To improve the mechanical performance of Mg-Al system alloys at high temperature, heat-resistant HPDC Mg alloys were developed as the ZA series, AX/J series, and AE series. The most representative heat-resistant HPDC Mg alloys are AE44 and MRI230 (Mg-6Al-2Ca-0.5Sr, wt%) alloys, which are used in the automobile industry. Then, we defined their strength–ductility synergy as the satisfactory level, as marked by a green, dotted line in Figure 1, where the best strength–ductility properties of AE44 and MRI230 alloys were selected. It is obvious that only a dozen of the alloys had better strength–ductility synergy than the satisfactory level, and they were mainly from the AE system, except one Mg-RE-based alloy (Mg-3La-5Gd), one Mg-RE-Zn-based alloy (Mg-4Zn-2La-3Y, ZLaW423), and one thin-wall Mg-Zn-Al-based alloys (Mg-8Zn-8Al, ZA88). With respect to HPDC Mg alloys, the mechanical performance is highly reliant on microstructures, particularly intermetallic phases on grain boundaries, which are mainly dependent on chemical compositions. Table 1 summarizes the average grain size (or cell size) and the dominant intermetallic phases along with their morphologies of representative HPDC Mg alloys with high strength and high ductility, such as those located in the area on the green, dotted lines in Figure 1. It is obvious that the intermetallic phases were various even in the same Mg-Al-RE system due to the different RE elements or other non-RE element additions. In the following, their microstructural characteristics are systematically discussed in detail.

Herein, the general measurements/analysis methods are simply introduced. Firstly, the main chemical compositions of the alloys were determined by inductively coupled plasma atomic emission spectroscopy (ICP-AES). Then, optical microscopy (OM) and backscatter scanning electron microscopy (SEM) were widely adopted to identify the grain (or cell) size and to observe the morphologies and distribution of the grains and intermetallic phases. Of course, the grains can be more accurately characterized using the electron back-scattered diffraction (EBSD) technique, and the types of intermetallic phases can be roughly estimated from SEM observations. The crystal structures of the various intermetallic phases along with dislocations were identified by employing X-ray diffraction (XRD) analysis and transmission electron microscopy (TEM), including TEM or scanning TEM (STEM) imaging, selected area electron diffraction (SAED) pattern analysis, energy-dispersive X-ray spectroscope (EDS) analysis, and two-beam bright-field (TBBF) or two-beam dark-field (TBDF) imaging techniques. Furthermore, the inner faults were analyzed using high-resolution TEM (HRTEM) analysis. Finally, the atomic resolution analysis was generally built on Cs-corrected high-angle annular dark-field scanning TEM (HAADF-STEM) imaging.

## 3. Microstructural Characteristics of AE Series Alloys

### 3.1. Mg-Al-MM Alloys

Misch-metal (MM) is widely adopted in traditional AE series alloys, and the representative alloys are AE42 and AE44. Zhu et al. [34] thoroughly examined the microstructure of HPDC AE42 alloy and revealed it consisted of primary α-Mg dendrites and intermetallic phases at the grain boundaries, including Al_11_RE_3_, Al_2_RE, and minor Mg_17_Al_12_. In the AE44 alloy, there are mainly two types of intermetallic phases: lamellae and particular [44,45], such as shown in Figure 2a. Many selected area electron diffraction (SAED) patterns indicated these two phases as Al_11_RE_3_ (body-centered orthorhombic structure, *a* = 0.4431 nm, *b* = 1.3142 nm, and *c* = 1.0130 nm [45,106]) and Al_2_RE (diamond structure, *a* = 0.8052 nm [106]), respectively (Figure 2b–d). Both phases can effectively impede dislocation motion and grain boundary migration. Different from the Mg_17_Al_12_ (*a* = 1.05 nm [107]) phase, both of them have high thermal stability. Thus, these two intermetallic phases contributed to the alloys’ high strength at both room temperature and high temperatures. Due to the greater addition of MM, more Al solute atoms in the AE44 alloy were consumed to form intermetallic phases. Therefore, it contains almost free Mg_17_Al_12_ and a higher volume of intermetallic particles at the grain boundaries, thus higher strength than the AE42 alloy. In addition, some metastable Al-RE intermetallic phases such as Al_2.12_RE_0.88_ (hexagonal *a* = 0.4478 nm and *c* = 0.4347 nm [45,106]) were reported in the HPDC AE44 alloy, such as shown in Figure 2e.

### 3.2. Influence of Other Alloying Elements on AE-Based Alloys

Usually, minor Mn (generally, ≤0.4 wt%) is added to Mg-Al-based alloys to decrease the concentration of Fe, thus improving the alloys’ corrosion resistance [108,109]. Recently, the influence of minor Mn on the microstructural and mechanical properties of HPDC AE44 alloy was studied [78,79]. It was found that Mn addition can improve the aging response and creep resistance of AE 44 alloy, although with no discernable changes to the dominant intermetallic phases. However, it leads to a high density of nanoscale Mn-containing precipitates (Figure 2f). Additionally, Mn addition inevitably results in Mn-containing intermetallic phases [110,111], and the most frequent one is Al_10_RE_2_Mn_7_ (rhombohedral structure, *a* = 0.904 nm and *c* = 1.317 nm [112]). Yang et al. [110] carefully examined the crystal structure of the Al_10_RE_2_Mn_7_ phase in AE44 alloy on the basis of SAED pattern inspections and high-resolution transmission electron microscopy (HRTEM) observations (Figure 3). They found many special inner-faults and an orientation relationship (OR) with another Mn-containing phase, Al_8_REMn_4_ (tetragonal structure, *a* = 0.898 nm and *c* = 0.517 nm [113]) as [010]_Al_8_REMn_4__//[-1101]_Al_10_RE_2_Mn_7__ and (101)_Al_8_REMn_4__//(11-20)_Al_10_RE_2_Mn_7__. Combined with density functional theory (DFT) calculations, it was concluded that the Al_10_RE_2_Mn_7_ phase could be transferred from the Al_8_REMn_4_ phase following the above OR, and the profuse inner-faults in the Al_10_RE_2_Mn_7_ phase was attributed to the blurry regions at ITBs in the Al_8_REMn_4_ phase [110].

In addition, adding some other alloying elements, such as Sr [80,100,101,102], Ca [68,81,100], Zn [36,37], and B [82], were adopted in the AE-based alloys. A minor Sr addition such as less than 0.5 wt% would not alter the crystal structure of the dominant intermetallic phases, but it alters the morphology of Al_11_RE_3_ from a lamellae shape or needle-like shape to a plate shape [80], such as shown in Figure 4a. The underlying cause is that the pre-existing tiny Al_4_Sr (body-centered tetragonal structure, *a* = 0.446 nm and *c* = 1.107 nm [114]) acted as nucleation sites for the Al_11_RE_3_ phase during solidification. More Sr additions such as 1 wt% into AE42 alloy would clearly change the intermetallic components at the grain boundaries (Figure 4b,c), resulting in the formation of blocky Mg_8_Al_4_Sr (hexagonal structure, *a* = 0.597 nm and *c* = 0.514 nm [102]). It is reported that the addition of Sr into Mg-Al-Ca-based alloys would change the Al-rich (Mg,Al)_2_Ca phase to the Mg-rich Mg_17_Sr_2_ phase, thus improving alloy’s strength and creep resistance [102]. Thus, Sr addition into different alloy systems has different effects.

Similarly, Ca additions into AE-based alloys also generates different modifications to microstructures. Qin et al. [81] reported that 1 wt% Ca addition into a Mg-4Al-3La (ALa43) alloy did not change the morphology of Al_11_RE_3_ and only introduced a few of the (Mg,Al)_2_Ca phase (dihexagonal C36 crystal structure, *a* = 0.594 nm, and *c* = 1.971 nm [37,38]) at the grain boundaries. Thus, the TYS of ALaX431 is comparative to the traditional ALa44 alloy. However, Ca addition resulted in many nanoscale Al_2_Ca precipitates (face-centered cubic-based C15 structure, *a* = 0.8022 nm [115,116]) in the matrix after heat treatment, such as aging at 200 °C. These precipitates significantly improve the TYS of the HPDC ALaX431 alloy by ~42 MPa [81]. More Ca addition such as 2 wt% into AE-based alloys would lead to many (Mg,Al)_2_Ca particles at the grain boundaries, forming a continuous or semicontinuous intermetallic skeleton (Figure 4d). This directly contributes to the alloy’s high TYS over 200 MPa but having poor ductility [68]. Additionally, 2 wt% Ca addition leads to the intermetallic components being more complex (Al_11_RE_3_, Al_8_RE_3_, Al_2.12_RE_0.88_, and Al_10_RE_2_Mn_7_), and the rod-like Al_8_RE_3_ phase with an unknown structure was firstly found in the AE-based alloys. Moreover, Ca addition obviously decreased the stability of the well-known Al_11_La_3_ phase, which finally decomposed into Al_2_(Ca,La) [117]. The TEM characterizations showed this process as Ca firstly segregated at the Mg/Al_11_La_3_ interface, then Al and Ca simultaneously segregated at the interface to form the Al_2_Ca phase, and finally the Al_11_La_3_ gradually decomposed into an Al_2_(Ca,La) structure. The DFT calculations revealed that the underlying cause was Ca atoms substituting for La atoms, decreasing the stability of Al_11_La_3_.

Zn addition into the conventional AE44 alloy introduces lamellae Al_2_REZn_2_ particles [52,53], but this phase has flat strengthening effects on the alloy’s strength and ductility. Inversely, it clearly decreases the creep resistance of the AE44 alloy even at moderate temperatures, such as 150 and 200 °C [53,118]. Moreover, Yang et al. [82] reported that trace B addition disturbed the intermetallic particles in the eutectic regions of ALa44 alloys and changed their morphology from a needle-like shape to a plate-like shape. However, this change was not friendly to the alloys’ strength.

### 3.3. Mg-Al-La Alloys

La is a cheap RE and has been widely adopted in the recent development of high-performance HPDC Mg alloys. Zhang et al. [83] investigated the influence of different La additions on the microstructures and mechanical properties of the HPDC Mg-4Al-0.3Mn alloy. They pointed out that La addition can refine grains and lead to the formation of Al_11_La_3_ (body-centered orthorhombic structure, *a* = 0.443 nm, *b* = 1.314 nm, and *c* = 1.013 nm [119,120]). As a result, the TYS monotonously improved as the La addition increased from 0 to 6 wt% not only at room temperature but also at high temperatures. In the ALa44 alloy, there were almost free Al_2_La particles. Recent work indicates that the dominant intermetallic phase is highly related to the Al/La ratio in Mg-Al-La-based alloys, and a new intermetallic compound with Al/La being ~3 was found in both gravity-cast and HPDC Mg-Al-La alloys [66,121]. It owns a highly similar morphology to the needle-like Al_11_La_3_ particles but results to significantly different diffraction peaks in XRD and SAED patterns. Furthermore, its atomic stacking sequences were also different from those of Al_11_La_3_. Wong et al. [99] reported it as a new (Al,Mg)_3_La phase (C-centered orthorhombic structure, *a* = 0.43365 nm, *b* = 1.88674 nm, and *c* = 0.4242 nm). However, the simulated electron diffraction patterns based on the provided crystal structure deviated from the experimentally observed SAED patterns in many cases (Figure 5a–c). Meng et al. [66] reconstructed the three-dimensional reciprocal lattice of the new phase (named η-Al_3_La) based on a series of continuous-tilting SAED patterns as a monoclinic structure with *a* = 0.4437 nm, *b* = 0.4508 nm, *c* = 0.9772 nm, and *β* = 103.5° and gave the crystal configuration based on various simulations. In addition, the authors found that the η-Al_3_La phase followed an OR in many cases, with the matrix as (011)_η_//(11-22)_Mg_, [106]_η_ deviated by 5.7–10.4° from [10-10]_Mg_. This demonstrates that the (001)_η_ plane was parallel to the (11-22)_Mg_ plane, thus being able to assist the pyramidal <**c** + **a**> dislocations, bypassing the η-Al_3_La particles. Hence, this is beneficial to achieving high strength and high ductility [66]. This new phase also exists in gravity-casting Mg-Al-La-based alloys and can provided efficient nucleation sites for some other intermetallic phases, such as ε-Mg_3_Ag (or ε-Mg_26−x_Ag_7+x_) and β-Mg_2_(Al, Ag)_3_ [99], following crystallographic ORs, such as (001)_ε_//(001)_η_, [010]_ε_//[010]_η_, (110)_β_//(001)_η_, and [-332]_β_//[010]_η_, respectively [122].

Zhang et al. [83,84] stated that the Al_11_RE_3_ (RE = La, Ce, Pr, and Nd) phase is metastable and decomposes into Al_2_RE when treated at relatively high temperatures. To examine the thermal stability of Al-La binary compounds, Yang et al. [124] conducted systematic first-principles calculations and indicated that in conventional Al-La binary compounds, only the Al_4_La (*I4/mmm*), Al_4_La (*Imm2*), and AlLa_3_ (*Pm-3m*) phases are metastable, and the Al_11_La_3_ phase has high stability at temperatures lower than 1000 K. Furthermore, Lv et al. [125] thoroughly examined the thermodynamic stability of all Al_11_RE_3_ intermetallic compounds compared with Al_2_RE + Al at a temperature range of 0–1000 K, using first principles calculations. It was found that the Al_11_RE_3_ phases with RE = La, Ce, Pr, Nd, Pm, Sm, Eu, Gd, and Tb are highly stable, while those with RE = Ho, Er, Tm, Lu, Y, and Sc are always unstable compared with the Al + Al_2_RE two-phase equilibrium. This viewpoint was proved to be experimental by previously reported work [123,126]. With respect to the newly found η-Al_3_La phase, the experimental examinations illustrated that it is metastable and partially transforms to α-Al_11_La_3_ during heat treatment, although with indiscernible changes to the particle morphologies [123]. Two distinct ORs between the parent and product phases were revealed based on REM observations: OR1—(001)_α_//(001)_η_ and [106]_α_//[010]_η_ (Figure 5d–f) and OR2—(001)_α_//(001)_η_ and [106]_α_//[106]_η_. According to atomic resolution scanning TEM (STEM) inspections, the α-Al_11_La_3_ to η-Al_3_La transformation is highly related to the pre-existing planar faults similar to the orientation twins. Therefore, an intermediate ε-Al_2.12_La_0.88_ phase was frequently observed during phase transformation, resulting in a two-stage path of η → ε → α. The corresponding DFT calculations indicated that the two-stage path is more energetically economical than the direct η → α path. Although this phase transformation can slightly improve the TYS, it clearly decreased the creep resistance at least at 200 and 90 MPa.

### 3.4. Mg-Al-Ce Alloys

Different REs in AE44-based alloys result in different microstructures. Zhang et al. [95] studied various Ce additions to Mg-4Al-0.4Mn alloy and reported that Ce has highly similar effects on the microstructures and mechanical properties of La (Figure 6a,b). The TYS of the studied alloys was monotonously improved as the Ce addition increased from 0 to 6 wt% at both room temperature and high temperatures. The ACe44 (Mg-4Al-4Ce, wt%) alloy owned satisfying mechanical properties and contained many blocky intermetallic particles, which were identified as Al_2_Ce [84,127]. In the Mg-Al-Ce system, the η-Al_3_Ce structured compound has hitherto not been reported. Since their mechanical properties were generally lower than ALa44, ACeLa44, or even AE44 alloys, limited attention was paid to ACe44-based alloys in the later investigations.

### 3.5. Mg-Al-Pr Alloy

Due to the exorbitant price, Pr is rarely adopted in preparing structural Mg alloys. In 2009, Zhang et al. [96] thoroughly investigated the impact of various Pr additions on the microstructures and mechanical properties of HPDC Mg-4Al-0.4Mn alloy. Different from La and Ce additions, a great many blocky particles (Al_2_Pr [128]) were observed in all of the studied Mg-Al-Pr alloys, except a few of lamellae or needle-like particles (Figure 6c). As the Pr addition increased, more large blocky particles appeared in the matrix, and some new particles, such as with a petal-like morphology, were generated [96]. Unfortunately, their detailed crystal structures were not revealed. It was reported that the room-temperature strength of Mg-Al-Pr alloys is lower than that of Mg-Al-La, Mg-Al-Ce, and Mg-Al-Nd alloys, and their strength at 200 °C is also lower than that of Mg-Al-La and Mg-Al-Ce alloys. Thus, the latter development of HPDC AE-based alloys took no account of Pr addition.

### 3.6. Mg-Al-Nd Alloy

Similar to Pr, Nd is also rarely adopted in developing HPDC Mg alloys due to the fact of its high price. Again, Zhang et al. [94] studied the influence of various Nd additions (1, 2, 4, and 6 wt.%) on the microstructures and mechanical properties of HPDC Mg-4Al-0.4Mn alloy. Their results showed that the dominant intermetallic particles were blocky (Al_2_Nd [129]) when the addition was below 2 wt%. As the Nd content increased, more needle-like particles and some dendritic particles with a coarse surface (Al_11_Nd_3_) formed (Figure 6d). Greater Nd addition could simultaneously improve the alloys’ strength and corrosion resistance. Afterwards, Su et al. [85] further studied the stability of Al-Nd intermetallics in ANd44 (Mg-4Al-4Nd-0.2Mn) alloy. They found that Al_11_Nd_3_ transformed to Al_2_Nd at temperatures above 300 °C, thus decreasing the strength and corrosion resistance.

### 3.7. Mg-Al-Sm Alloys

Sm has a unique orthorhombic structure and owns a relatively great difference of the solid solubility in the Mg matrix as the temperature decreases, thus being able to improve the alloys’ strength by precipitation strengthening or solid-solution strengthening [130]. Yang et al. [86] thoroughly studied the intermetallic phases in an HPDC ASm44 (Mg-4Al-4Sm-0.3Mn, wt%) alloy and found that almost all of the intermetallic particles presented a petaloid morphology (Figure 6e). This petaloid phase contributed to the high strength–ductility synergy for the ASm44 alloy, with the TYS of 157 ± 3 MPa and elongation to fracture of 21.1 ± 2.6%. Detailed TEM observations indicated that the petaloid phase was not Al_11_Sm_3_ (body-centered orthorhombic structure, *a* = 0.4287 nm, *b* = 1.2761 nm, and *c* = 0.9905 nm [131]) but Al_2_Sm (face-centered cubic structure, *a* = 0.7945 nm [132]) and composed of multiple (111) twins (Figure 7a–c). Many HRTEM observations showed that most of the twin boundaries featured a sandwiched structure with the filling being the Al_12_Sm_2_Mn_5_ phase. These peculiar Al_2_Sm structures were related to the random heterogeneous nucleation of Al_2_Sm particles on the pre-existing Al_12_Sm_2_Mn_5_ sites during solidification. As the Sm addition increased to its maximum solid solubility in Mg (~5.6 wt.% [130]), some needle-like and dendritic particles appeared at grain boundaries [87]. Yang et al. [87] carefully examined the crystal structures of the intermetallic phases in ASm46 (Mg-4Al-5.6Sm-0.3Mn, wt%), and it is very interesting that the Al_2_Sm phase simultaneously presented needle-like (Figure 7d,e), blocky, and petaloid morphologies, while the Al_11_Sm_3_ phase was dendritic (Figure 7f,g). In addition, some blocky Al_2.12_Sm_0.88_ particles were detected at the grain boundaries. Furthermore, the strength of the ASm46 alloy could be improved by aging, with the increments being as high as ~22 MPa at room temperature. TEM examinations (Figure 8) indicated that dense nanoscale precipitates precipitated in the Mg matrix, with an average size of ~3.7 nm. Atomic resolution STEM observations revealed these precipitates as Al_3_Sm (hexagonal structure, a = 0.6380 nm and c = 0.4597 nm [133]), following an OR with the Mg matrix, such as (202¯1)_Al_3_Sm_//(11¯01)_Mg_ and [12¯10]_Al_3_Sm_//[12¯10]_Mg_. The strengthening mechanism of these precipitates was identified based on TEM examinations as Orowan dislocation bypassing.

### 3.8. Mg-Al-Gd Alloys

Gd is one of the most frequently used RE elements in wrought Mg-RE-based alloys, because it has a very high solid solubility in Mg and is widely accepted as being favorable for improving alloys’ plasticity and strength [134,135,136]. Qin et al. [88] studied the microstructural and mechanical properties of HPDC AGd44 (Mg-4Al-4Gd-0.3Mn, wt%). Different from the AE44-based alloys with other RE elements, as stated in the above, there were much fewer intermetallic particles in the HPDC AGd44 alloy, and only blocky and petaloid particles were observed. After aging at 200 °C for 48 h, there were no discernible changes to the pre-existing intermetallic phases, while some fine lath-like precipitates with relatively low contrast formed at the grain boundaries. The TYS of the AG44 alloy was improved by ~30 MPa. TEM examinations showed that most of the blocky particles were Al_2_Gd (face-centered cubic structure, *a* = 0.79 nm [129,137]), while the lath-like precipitate was Mg_17_Al_12_ following the reported Pitsch–Schrader OR or Burgers OR with the Mg matrix [138]. The blocky Al_2_Gd particles consisted of several parts, and each part possibly contained profuse planar faults such as twin boundaries (Figure 9a–c). In addition, the twinned Al_12_Gd_2_Mn_5_ domains were frequently found embedded in Al_2_Gd phases. Petaloid particles were also observed in the AGd44 alloy, but its petals were different from those of the petaloid Al_2_Sm in which center axes are found and are round by amounts of small Al_2_Gd cells (Figure 9d–f).

### 3.9. Mg-Al-La-Sm/Gd Alloys

As mentioned above, adding different RE elements results in a distinctive influence on the microstructural and mechanical properties of HPDC Mg–Al-based alloys. Considering the blistering that usually occurs during heat treatments, solid-solution strengthening is ordinarily ignored in HPDC Mg alloys, and the strengthening mainly relies on the intermetallic phase at cell boundaries, except fine grains [30,31,139,140]. However, short-term aging treatment at relatively lower temperatures is also expected to assist and has been successfully adopted in some HPDC alloys [27,87,88]. Yang et al. [87] reported that ASm46 alloy has excellent aging hardening by forming many nanoscale Al_3_Sm precipitates, which suggests some Sm solute atoms in the Mg matrix, thus probably extra solid-solution strengthening in traditional HPDC Mg alloys. Then, Yang et al. [69] studied the microstructural and mechanical properties of HPDC ALaSm432 (Mg-4Al-3La-2Sm-0.3Mn, wt%) alloy. The intermetallic phases become diversified with many particles with new morphologies appearing, but only the hammer-like Al_2_RE particles were not observed in traditional Mg-Al-RE-based alloys. Their morphologies are dependent on the phase chemical compositions, and more Sm-containing particles own similar morphologies to those in ASm44 or ASm46 alloys, while more La-containing particles similar to those in ALa44 or ALaM440 alloys. As a result, ALaSm432 alloy has lower ductility than ASm44 alloy [86] and comparable strength with ALaM440 and ASm46 alloys [66,87].

Recently, Lv et al. [89] investigated the effects of 2 wt% Gd addition on the microstructural and mechanical properties of an HPDC ALa43 (Mg-4Al-3La-0.3Mn, wt%) alloy, namely ALaGd432 herein. As indicated in Figure 1, the HPDC ALaGd432 alloy had outstanding strength–ductility synergy at room temperature, with TYS of 181 MPa and elongation to fracture of ~14%. The microstructure of the ALaGd432 alloy was clearly modified, and the intermetallic phases become much more complex than Sm addition. The dominant needle-like and blocky phases were η-Al_3_(La,Gd) and Al_2_(Gd,La), respectively, which were enriched with La and Gd, respectively (Figure 10a–d). The well-reported planar faults in Refs. [86,123] were found in both the η-Al_3_(La,Gd) and Al_2_(Gd,La) phases. Additionally, three new intermetallic phases were observed. The first one was the claviform phase, which was ~300 nm in diameter and 0.8–6 μm in length (Figure 10e). This phase was identified based on SAED inspection as Al_7_(La,Gd)_3_, which owns an AlB_2_ structure with the experimentally calculated lattice parameters of *a* = 0.458 nm and *c* = 0.375 nm. The second one was a flaky η-Al_3_(La,Gd) phase (Figure 10f), and the last one was fine Al_3_Gd_2_ particles, which possibly own a simple cubic structure with *a* = 1.647 nm and follows an OR with the matrix as (130)_Al_3_Gd_2__//(0001)_Mg_, [3-1-2]_Al_3_Gd_2__//[-2110]_Mg_ (Figure 10g). Considering the minor Al_2.12_(La,Gd)_0.88_ and the hammer-like Al_2_(Gd,La) phase, there were seven kinds of intermetallic phases. The multiscale and multityped intermetallic particles contributed to the outstanding mechanical properties of the HPDC ALaGd432 alloy.

## 4. Microstructural Characteristics of Mg-RE-Based Alloys

### 4.1. Mg-RE Binary Alloy

Mg-RE-based alloys have shown great potential for powertrain applications due to the fact of their outstanding yield strength at room temperature and excellent high-temperature performance [71,73,77,103,141,142,143,144,145,146]. The benchmarks are the MEZ (La-, Ce-, and/or Nd-rich Mg-2.5Nd-0.3Zn-0.3Mn) and AM-HP2+ (Mg-1.7La-1.0Ce-1.0Nd-0.45Zn) alloys [77,144,145]. In HPDC Mg-RE-based alloys, the RE content is usually lower than 4 wt%, and a higher RE would lead to serious brittleness [144,146]. Zhu et al. [147] studied the microstructures of three Mg-RE binary alloys: Mg-3.44La, Mg-2.87Ce, and Mg-2.60Nd (Figure 11). It is obvious that the Mg-3.44La alloy had the greatest number of intermetallic particles at the grain boundaries, while the Mg-2.60Nd alloys had the lowest. In addition, the grains in the Mg-3.44La alloy were the finest, and those in the Mg-2.60Nd alloy were the coarsest. The morphologies of the intermetallic particles gradually changed from net-work to lamellae following La, Ce to Nd. Except for the various RE contents, the main underlying cause was that the different RE elements generated different intermetallic phases: Mg_12_La, Mg_12_Ce and Mg_3_Nd in the Mg-3.44La, Mg-2.87Ce and Mg-2.60Nd alloys. In addition, the RE solute levels in the matrix of these three alloys were different. After heat treatment, approximately free precipitates were observed in the Mg-3.44La alloy, while a few precipitates in the Mg-2.87Ce alloy and a high density of β’ (Mg_3_Nd, orthorhombic structure, *a* = 0.64 nm, *b* = 1.14 nm and *c* = 0.52 nm [91]) precipitates. This indicates that the best thermal stability was for the Mg-3.44La alloy.

### 4.2. Mg-RE Ternary Alloy

Furthermore, a Mg-La system has better castability than other Mg-RE systems [103]. As a result, it was widely selected for the development of high-performance HPDC alloys, such as by further alloying with other RE elements with relatively high solubility in Mg, e.g., Y, Nd, and Gd [91]. The experimental results indicated that further RE alloying can significantly improve an alloy’s TYS, particularly after T6 treatment. For example, the TYS of the HPDC Mg-2.8La alloy was ~85 MPa and that of the Mg-2.8La-5.6Gd alloy ~140 MPa, which was interestingly improved to ~210 MPa after T6 treatment. TEM examinations showed a few precipitates in the matrix after creeping, approximately free precipitate after T4 treatment while dense precipitates after T6 treatment. Similar precipitate aggregating in lines that generally appeared in the crept sample was also observed in the T6 sample, and the fine precipitates were identified as the conventional peak-aging precipitate, β’.

### 4.3. Zn Addition into Mg-RE Alloy

Easton et al. [142] reported that more than 0.5 wt% Zn addition into Mg-Nd-based alloys can improve the mechanical properties, but it also clearly increases the hot tearing susceptibility. However, Choudhuri et al. [148] added 0.8 wt% Zn into an HPDC Mg-3.6Nd-2.3La alloy and found that Zn addition markedly improves an alloy’s creep resistance properties. In addition, they attributed this to the fact that Zn lead to the formation of γ” (ordered hexagonal structure, *a* = 0.56 nm and *c* = 0.45 nm [10]) precipitates during creep. Recently, Hua et al. [46] studied the microstructural and mechanical properties of an HPDC Mg-2La-3Y-0.3Mn alloy with 4 wt% Zn addition. Indeed, this alloy was very brittle, and only the compressive strength and ductility were reported, as listed in Figure 1. However, it has to be noted that its TYS was very high, such as ~202 MPa, which is comparable with that of the AEX422 alloy [68]. The 4 wt% Zn addition changed the morphologies of the grain boundary particles, leading to the dominant intermetallic phases irregularly aggregating together (Figure 12a). TEM examinations (Figure 12b–e) identified that the relatively dark intermetallic phase was Mg_12_RE (body-centered tetragonal structure, *a* =1.052 nm and *c* =0.596 nm [149]) and the bright one was the W phase (face-centered cubic structure, *a* = 0.684 nm [150,151]). Moreover, a few of LPSO plates were found in this alloy (Figure 12f).

### 4.4. Al Addition into Mg-RE Alloy

As discussed above, the AE system is the most representative of the HPDC alloys with high strength–ductility synergy, while the Mg-RE system has great potential in obtaining high-strength, although with limited ductility. Thus, adding minor Al into Mg-RE-based alloys might have impressive influence on the mechanical properties. For example, Dong et al. [93] studied minor (0.5 wt%) Al addition on the mechanical properties and microstructures of an HPDC Mg-RE (Mg-3.5RE(La,Ce,Nd)-1.5Gd-0.3Mn, wt%) alloy. Their results showed that minor Al addition observably modified the alloy’s ductility, from less than 1% up to ~2.5%. More interestingly, minor Al significantly decreased the creep resistance at 300 °C. Microstructural characteristics (Figure 13a,b) showed that minor Al addition introduced some Al_2_RE_3_ (tetragonal structure, *a* = 0.8344 nm, *c* = 0.7656 nm [152]) embedded in the Mg1_2_RE phase at grain boundaries. After creep, many fine Al_2_Gd plates were precipitated in the matrix (Figure 13b). Furthermore, some AlMnGd short-range clusters (Figure 13c) were firstly reported in HPDC Mg-RE-Al-based alloys, and they had relatively high thermal stability and would generate during creep. Recently, Rong et al. [71] investigated the effects of 1wt% (and 2 wt%) Al contents on the microstructural characteristics and properties of HPDC thin-wall Mg-3RE-0.5Zn alloy. Moreover, they proved that Al addition can clearly increase an alloy’s ductility and simultaneously slightly improve the alloy’s strength. However, more Al addition led to the formation of the Al_11_RE_3_ and Al_2_RE phases not Al_2_RE_3_. Finally, Al addition clearly decreased the thermal conductivity of the Mg-RE-based alloys.

### 4.5. Zn + Al Addition into Mg-RE Alloy

Bai et al. [153,154] developed an HPDC Mg-6Y-3Zn-1Al (WZA631) alloy which can be regarded as Zn + Al addition into a Mg-6Y alloy. The majority of the WZA631 alloy consisted of equiaxed Mg grains, and massive and amorphous (Al,Zn)_2_Y phases (cubic structure, *a* = 0.76 nm) were firstly reported. Simultaneously, honeycomb 18R-LPSO particles formed at the grain boundaries. This alloy exhibited an excellent strength–ductility synergy with a TYS of ~175 MPa and an elongation of ~9.8%. Afterwards, the authors studied the thermal stability of the studied alloys. However, the aging hardening response of this alloys was not involved in their work.

## 5. Microstructural Characteristics of AX-Based Alloys

As indicated in Figure 1, Mg-Al-Ca/Sr-based alloys also exhibits satisfactory strength–ductility synergy. Generally, approximately continuous intermetallic skeletons distributed on grain boundaries in Mg-Al-Ca/Sr-based alloys [31,49,90,91,104,105]. In the AX53 alloy, the dominant intermetallic phases are C14-Mg_2_Ca (hexagonal structure, *a* = 0.622 nm and *c* = 1.010 nm [36]) and C36-(Mg,Al)_2_Ca, while those in the AJ53 alloy are Mg_17_Sr_2_ (hexagonal structure, *a* = 1.0533 nm and *c* = 1.0342 nm [90]) and Al_4_Sr. In developing heat-resistant Mg-Al-Ca-based alloys, minor Sr addition was frequently adopted due to the fact that Sr could increase the Al solute content in matrix. To date, three classic HPDC AX-based alloys were developed and have been applied in the automobile industry, namely, AXJ530, MRI230D, and MRI153M [104]. These three alloys have similar microstructures to each other, except that the intermetallic skeletons in the AXJ530 and MRI230D alloys are relatively more continuous and structured. As a result, the former two alloys have a relatively higher strength, while the latest one owns better ductility. It is reported that during heat treatments many nanoscale Al_2_Ca plates precipitate in the matrix of Mg alloys containing Al and Ca (Figure 14). As mentioned in Section 3.1, these fine precipitates can clearly improve an alloy’s strength, and this method was adopted in a recent paper dealing with the development an HPDC AX73 (Mg-7Al-3Ca, wt%) alloy [92]. Unfortunately, the TYS of AX73 alloy was only improved by ~12 MPa, and the weak aging hardening might be due to the fact of grain or pore coarsening during aging treatment.

## 6. Microstructural Characteristics of ZA-Based Alloys

The Mg-Zn-Al system with relatively high Zn content firstly reported in the 1970s as an alternative to the commonly used aluminum-rich alloy, has even better castability than commercial AZ-based alloys [155,156,157]. Furthermore, the icosahedral quasicrystal phase (I-phase) was frequently observed in Mg-Zn-Al-based alloys [65]. As is well known, an I-phase possessing five-fold symmetry and a quasiperiodic structure is well accepted to play an important role in alloys’ high strength and high ductility [155,158,159]. In 2001, Vogel et al. [157] studied the microstructure of a classic ZA85 (Mg-8Zn-5Al, wt%) and found that this alloy exhibited typical dendritic solidification structures (Figure 15a,b). SAED patterns indicated the grain-boundary intermetallic phase as I-phase (Figure 15c–e). Unfortunately, the TYS of the HPDC ZA-based alloys was generally lower than 170 MPa, and the ductility was extremely lower than 5% [75,160]. Recently, Yang et al. [65] developed an ultrathin-wall HPDC ZA88 (Mg-8Zn-8Al) alloy, and this alloy shows a standout TYS of ~248 MPa, although with a very low ductility of ~2.3%. TEM observations (Figure 16a,b) demonstrated an interesting bimodal microstructure that consisted of two kinds of Mg grains: relatively coarse α-Mg grains (~5.6 µm) and ultra-fine β-Mg grains (~0.67 µm). This phenomenon of two types of Mg grains is highly similar to that in traditional Mg-RE-based alloys, as mentioned above. However, the grain size of the ZA88 alloy was significantly finer than that in the Mg-RE-based alloys (Table 1), and the underlying reason might be the super cooling rate due to the use of then-wall HPDC technique. Furthermore, there were masses of granular intermetallic particles in eutectic regions, with the average size being ~51 nm (Figure 16c). EDS analysis (Figure 16d) showed that Zn and Al mainly aggregated in intermetallic particles, while Zn segregation near the intermetallic skeletons could be observed. Corresponding SAED inspection and STEM examinations revealed that the nanosized intermetallic particles were also I-phase but with obvious surrounding amorphous areas in which some Zn-rich clusters could be observed in some cases. Finally, Zn-rich clusters, in which atomic-columns visibly deviated from the hexagonal Mg sites due to the fact of Zn segregation or extra Zn-rich columns, and some plate-like pyramidal precipitates were observed in the matrix.

## 7. Summary and Outlook

It is well known that the mechanical properties of HPDC Mg alloys are highly reliant on the microstructural characteristics. Grain refinement can simultaneously improve alloys’ strength and ductility at room temperatures. However, the grain size of HPDC Mg alloys are comparable in a range of 5–20 μm (except thin-wall components), and there is no clear relationship between fine grain and high strength–ductility. Inversely, the HPDC Mg alloys with high strength exhibited different intermetallic characteristics than those in the alloys with high ductility. Among the reported HPDC Mg alloys, AE system alloys exhibit the most excellent strength–ductility synergy. Different intermetallic phases contribute to different strengthening effects and mechanisms, and the newly reported needle-like η-Al_3_La phase and the petaloid Al_2_Sm phase are proved having satisfactory effects on strength and ductility. In addition, the strength can be improved by aging in some AE alloys, such as with Sm and Gd, due to the precipitation of fine Al_3_Sm, Al_2_Gd, or Mg_17_Al_12_. Finally, further alloying could obviously improve the strength of AE-based alloys. Because of its high-strength, the Mg-Al-La system has been widely selected to be modified by alloying, such as Sm, Gd, Ca, and Sr. These methods could clearly improve an alloy’s strength, particularly alloying by Ca plus the following aging, and the underlying causes are based on changing the intermetallic phase components. It is unfortunate that the aging hardening responses of adding Sm, Gd, and Sr have not been investigated to date. On the other hand, further alloying of the Mg-Al-La system, particularly by Ca, would decrease the creep resistance at high temperature and might lead to abnormal creep stress exponents. In addition, Ca and Sr additions to AE-based alloys clearly decreases the thermal conductivity and seriously increases the tendency of hot cracking during HPDC. Therefore, further developing high-performance AE-based alloys should carefully pay attention to how the synergy between strength–ductility and creep resistance or other properties such as thermal conductivity and castability is coordinated.

Except for the AE system, the Mg-RE system should also be appreciated due to the fact of its outstanding high strength. Further increasing its ductility is the key to developing new HPDC alloys with high strength and high ductility. Recent work indicates that minor Al is prominent in enhancing the ductility of Mg-RE-based alloys, and Zn addition is beneficial for increasing an alloy’s strength but detrimental for the ductility and castability. Additionally, adding minor Al to Mg-RE-based alloys generates many fine Al_2_RE plates during heat treatment. Generally, a high density of fine precipitates is extremely helpful for enhancing the strength, but few investigations have been conducted to handle the aging hardening response of Al-modified Mg-RE-based alloys. Moreover, Mg-RE-based alloys ordinarily have excellent high-temperature properties and high thermal conductivity. Thus, Mg-RE system has the most potential for developing alloys with high strength, high creep resistance, and high thermal conductivity. Finally, the ZA system is also worthy to be investigated due to the fact of its low cost, high castability, and special intermetallic phases. Unfortunately, modifying ZA system alloys is rarely reported. This results in a poor understanding of the effects of adding other alloying elements, such as RE, Ca, Sr, and Si.

Modifying an alloy’s chemical compositions is the key to designing new HPDC alloys with high strength and high ductility, and the following heat treatments at relatively low temperatures were efficient for enhancing the alloys’ properties, particularly yield strength. However, aimed at promoting the wide application of HPDC Mg alloys, such as in the automobile and aerospace industries, other properties, such as creep resistance, thermal conductivity, corrosion resistance, and castability, should also be appreciated during future development. The combination property is much more important. RE-containing Mg alloys might be the most potential candidate for future wide application. Generally, complex intermetallic phases on the grain boundaries are expected to achieve moderate strength and high ductility, and a continuous or semicontinuous lamellae intermetallic skeleton is for ultrahigh strength. As mentioned in the above, micro-alloying and multi-alloying exhibit an excellent effect on modifying the microstructures of the RE-containing Mg alloys and should be paid attention to in future investigations.

## Figures and Tables

**Figure 1 materials-16-01954-f001:**
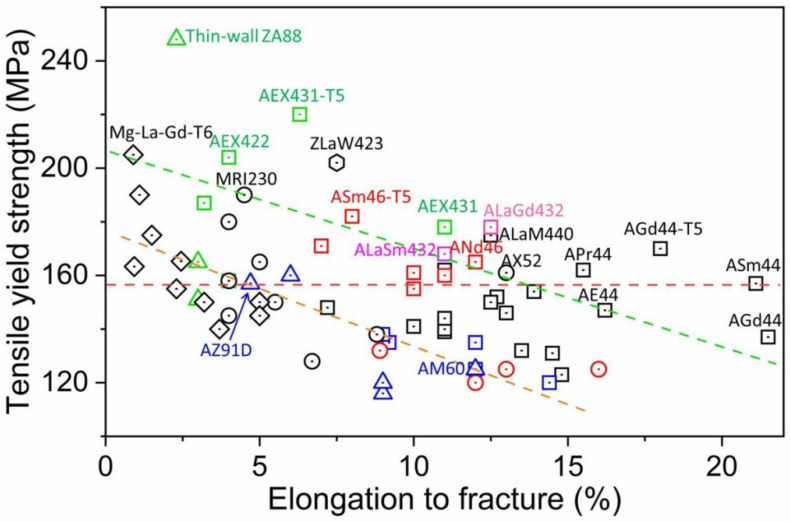
Room temperature TYS versus elongation of various HPDC Mg alloys.

**Figure 2 materials-16-01954-f002:**
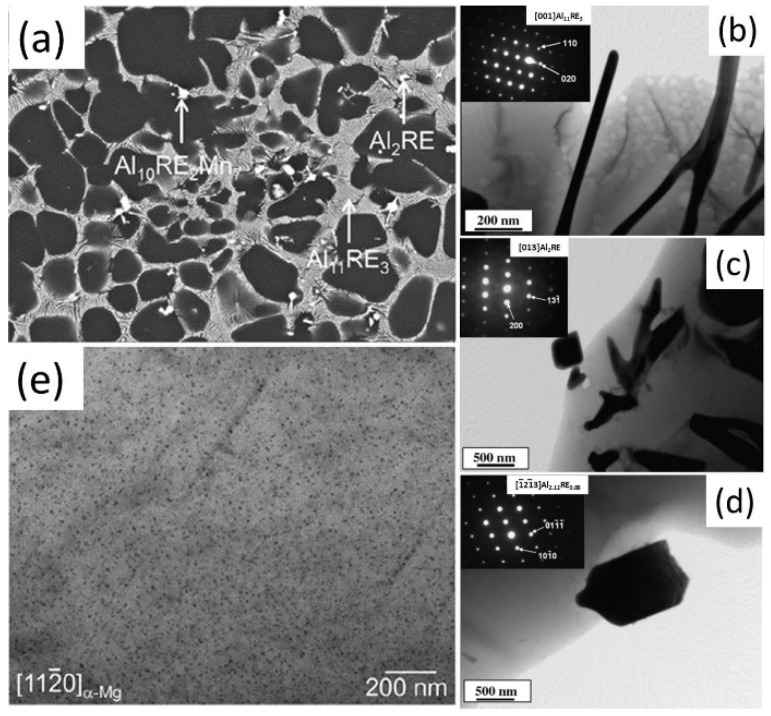
(**a**) Representative SEM image of traditional AE44 alloy, BF-TEM images, and/or SAED patterns of (**b**–**d**) intermetallic phases and (**e**) fine precipitates in the Mg matrix [50,106].

**Figure 3 materials-16-01954-f003:**
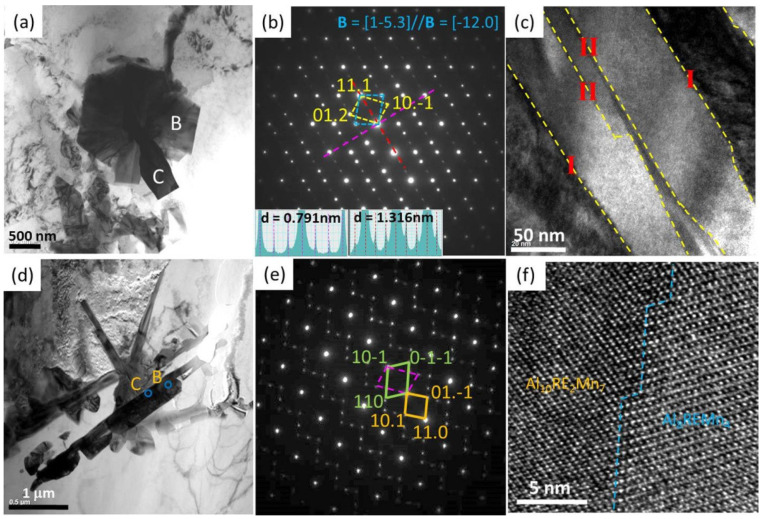
(**a**,**c**,**d**) BF-TEM images; (**b**,**e**) SAED patterns; (**f**) high-resolution TEM (HRTEM) images of Al-Mn-RE phases in AE44 alloys [110,111].

**Figure 4 materials-16-01954-f004:**
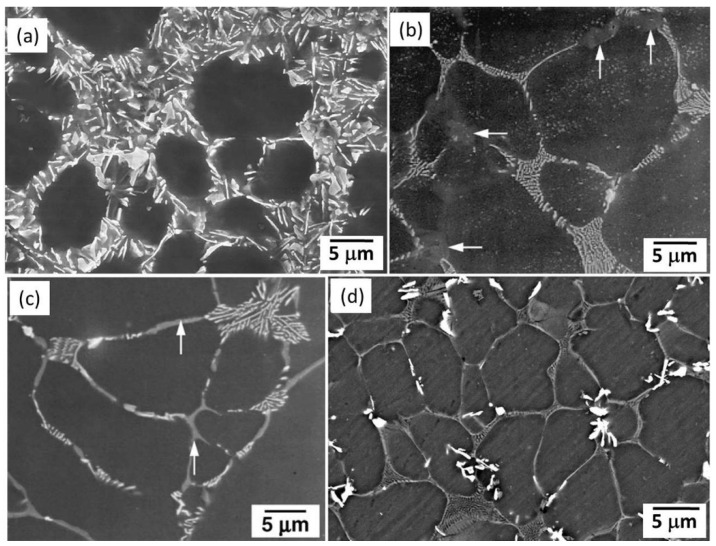
SEM images of the HPDC (**a**) ALa44 + 0.5Sr [80]; (**b**) AE42 + 0.5Sr [102]: (**c**) AE42 + 1.0 Sr [102]; (**d**) AE42 + 2.0Ca [68] alloys.

**Figure 5 materials-16-01954-f005:**
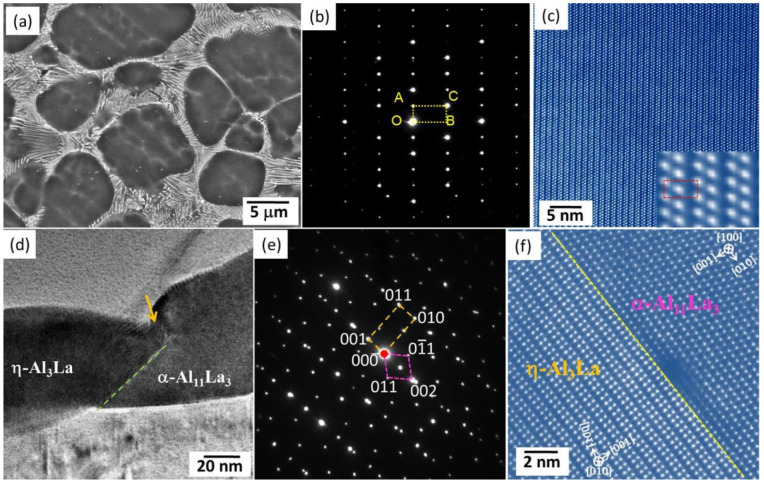
(**a**) SEM image; (**b**) SAED pattern; (**c**) atomic resolution HAADF-STEM image of the new phase in ALaM440 alloy; (**d**) TEM image; (**e**) SAED pattern; (**f**) atomic resolution HAADF-STEM image of the coexistence of η-Al_3_La and Al_11_La_3_ [66,123].

**Figure 6 materials-16-01954-f006:**
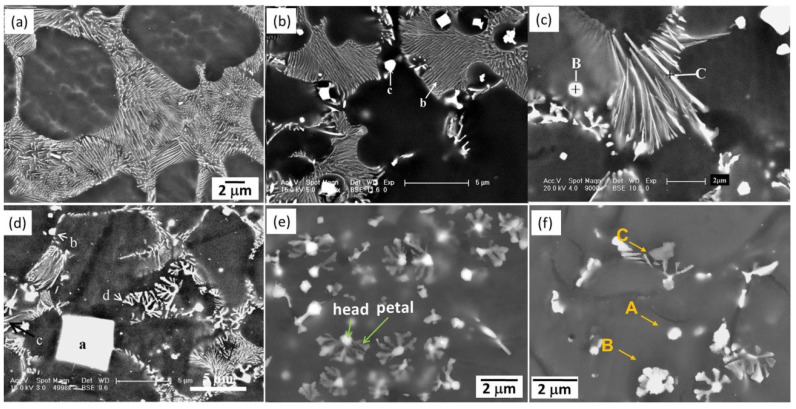
SEM images showing the microstructures of HPDC (**a**) ALa44; (**b**) ACe44 [95]; (**c**) APr44 [96]; (**d**) ANd44 [94]; (**e**) ASm44 [86]; (**f**) AGd44 [88] alloy.

**Figure 7 materials-16-01954-f007:**
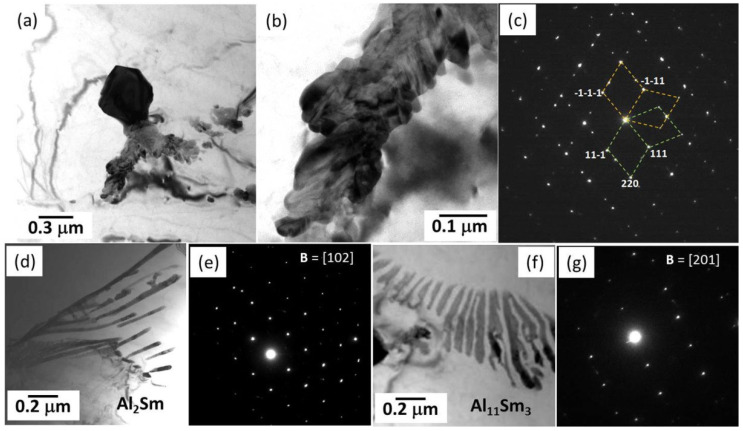
TEM images along with the corresponding SAED patterns of the intermetallic phases in Mg-Al-Sm-based alloys [86,87].

**Figure 8 materials-16-01954-f008:**
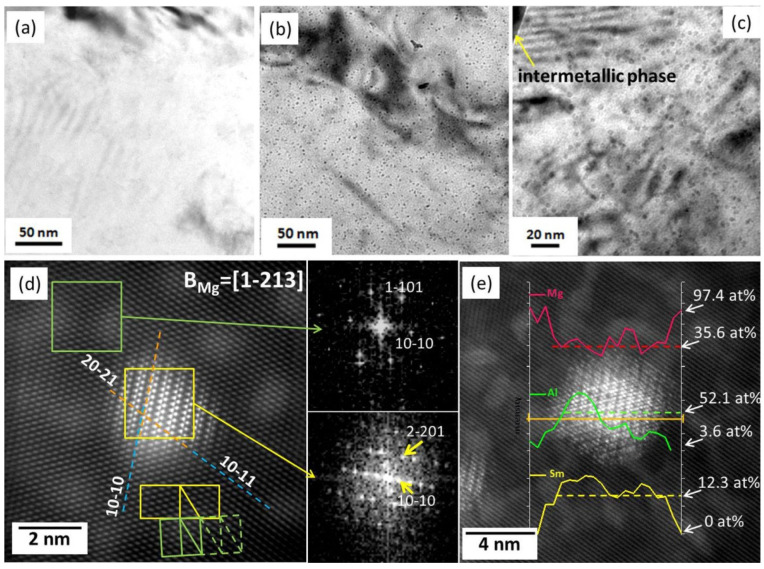
(**a**–**c**) TEM and (**d**,**e**) atomic-resolution STEM images showing precipitates in ASm46 alloy [87].

**Figure 9 materials-16-01954-f009:**
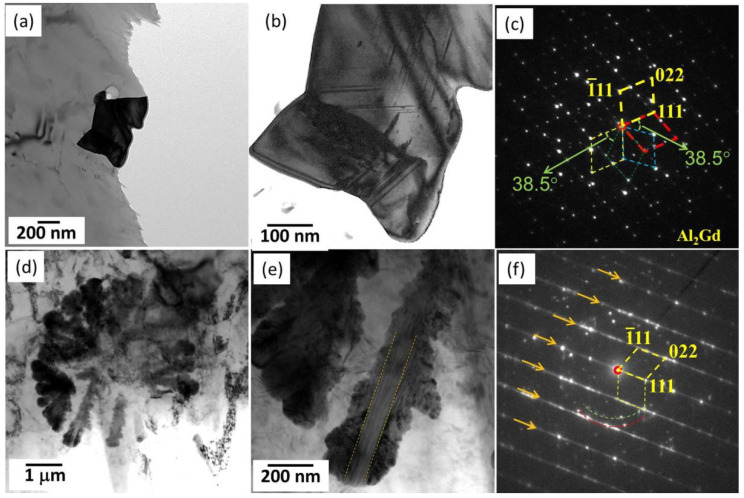
TEM images and the corresponding SAED patterns of Al_2_Gd in the AGd44 alloys [88].

**Figure 10 materials-16-01954-f010:**
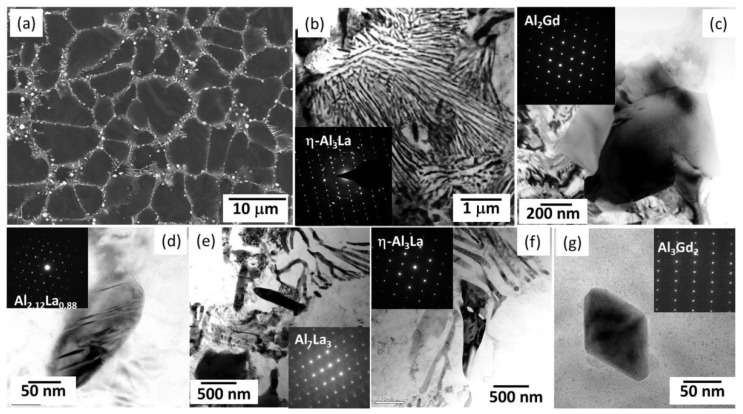
(**a**) SEM image of ALaGd432 alloy and (**b**–**g**)TEM images along with the corresponding SAED patterns (the inserts) of the intermetallic phases [89].

**Figure 11 materials-16-01954-f011:**
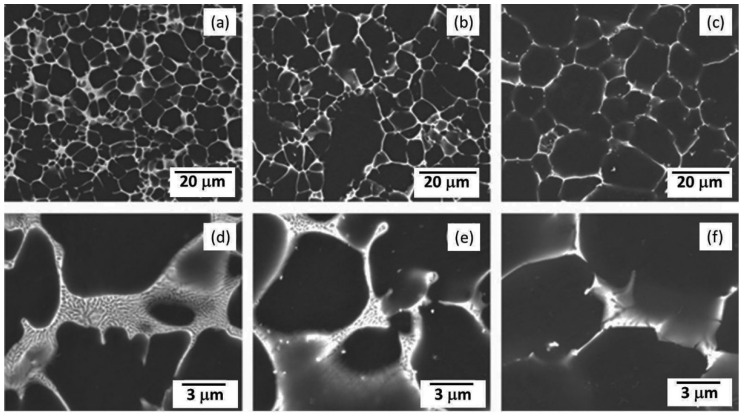
SEM images showing microstructures in the HPDC (**a**,**d**) Mg-3.44La; (**b**,**e**) Mg-2.87Ce; (**c**,**f**) Mg-2.60Nd alloys [147].

**Figure 12 materials-16-01954-f012:**
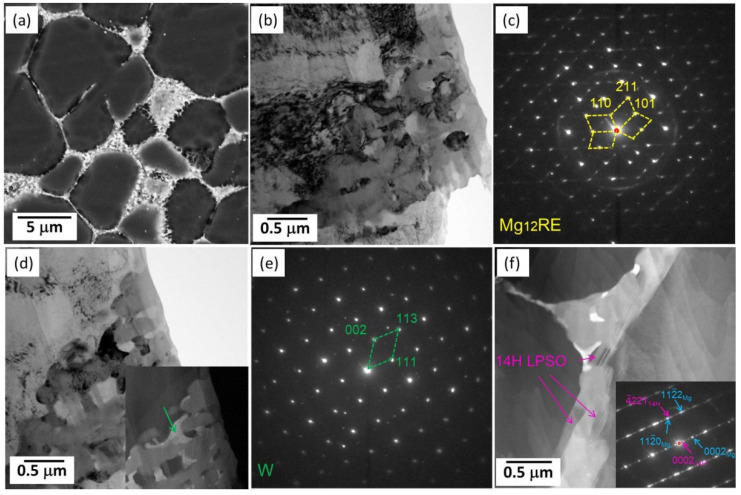
(**a**) SEM image of the HPDC ZLaW423 alloy, (**b**,**d**) TEM images along with (**c**,**e**) the corresponding SAED patterns of the dominant intermetallic phases, and (**f**) HAADF-STEM image showing LPSO plates with the corresponding SAED pattern shown in the insert [46].

**Figure 13 materials-16-01954-f013:**
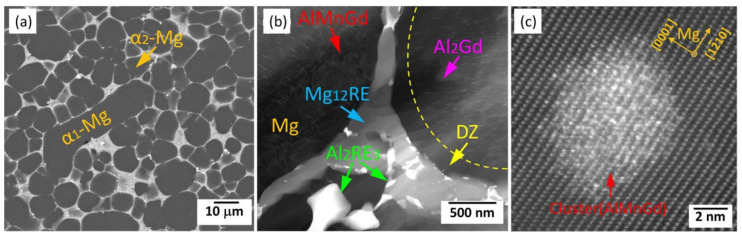
(**a**,**b**) SEM images of the aged Mg-3.5RE(La,Ce,Nd)-1.5Gd-0.5Al-0.3Mn sample and (**c**) an atomic-resolution HAADF-STEM image of AlMnGd cluster [93].

**Figure 14 materials-16-01954-f014:**
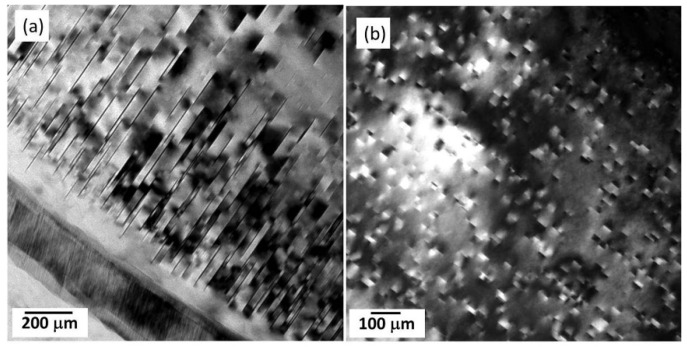
(**a**,**b**) TEM images showing fine plate-like precipitates in the heat-treated AEX422 alloys [118].

**Figure 15 materials-16-01954-f015:**
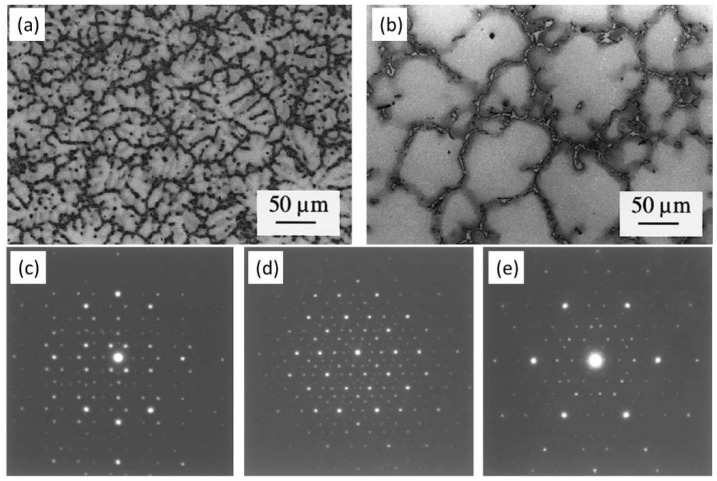
(**a**,**b**) OM images and (**c**–**e**) SAED patterns of intermetallic phases in the HPDC ZA85 alloy [157].

**Figure 16 materials-16-01954-f016:**
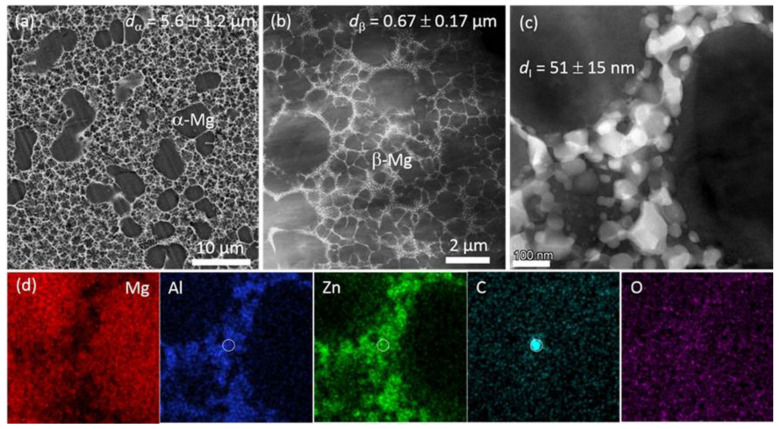
(**a**) SEM image; (**b**,**c**) HAADF-STEM images; (**d**) EDS mappings of the region shown in (**c**) in the HPDC ZA88 alloy [65].

**Table 1 materials-16-01954-t001:** Grain size and dominant intermetallic phases in various HPDC Mg alloys.

Alloy	Grain Size (μm)	Dominant Intermetallic Phases	Reference
AE44	~14	Acicular/needle-like Al_11_RE_3_, blocky Al_2_RE	[42,43]
ALa44	7–16	Acicular/needle-like Al_11_La_3_	[81,82]
ALaM440	~15.6	Acicular η-Al_3_La (or (Mg,Al)_3_La)	[64,97]
ACe44	~14	Acicular/needle-like Al_11_Ce_3_, blocky Al_2_Ce	[93]
APr44	~8.5	Quadrangle/polygon Al_2_Pr, acicular/petal-like Al_11_Pr_3_	[94]
ANd44	~10.5	Quadrangle Al_2_Nd, dendritic/coarse-surface Al_11_Nd_3_	[83,92]
ASm44	~9.6	Petaloid Al_2_Sm	[84]
ASm46	~6.3	Petaloid/blocky/short-rod-like Al_2_Sm, dendritic Al_11_Sm_3_	[85]
AGd44	~5.9	Petaloid/blocky Al_2_Gd	[86]
AE432	~6.1	Acicular/flaky η-Al_3_RE, blocky/spindle-shapedAl_2_RE, claviform Al_7_RE_3_	[67,87]
AEX series	~7	Lamellae (Mg,Al)_2_Ca, lath-like Al_11_RE_3_, rod-like η-Al_3_RE, blocky Al_2.12_RE_0.88_	[66,79,98]
AEJ series	~13.6	Blocky Al_4_Sr or Mg_8_Al_4_Sr, lamellae Mg_17_Sr_2_	[78,98,99,100]
ALa44+B	10.7	Acicular Al_11_RE_3_, blocky Al_2_RE	[80]
Mg-RE series	~5.1	Net-work Mg_12_RE, lamellae Mg_3_RE, plate-like β-series precipitates	[89,101]
ZLaW423	~6.1	Aggregated lamellae Mg_12_RE and blocky W phase, lath-like LPSO	[44]
Mg-RE-Al	α1 = 10–30, α2 = 2–10	Lamellae Mg_12_RE, blocky Al_2_RE_3_, precipitated AlMnRE cluster/Al_2_RE	[69,91]
AX/J series	~6.1	Lamellae (Mg,Al)_2_Ca/Mg_17_Sr_2_, net-work Mg_2_Ca/Al_4_Sr, precipitated Al_2_Ca	[29,47,88,89,90,102,103]
ZA88	α1 = ~5.6, α2 = ~0.67	Nanosized icosahedral quasicrystal phase	[63]

## Data Availability

No new data.

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
