# Peer review of "Microstructural Characteristics of High-Pressure Die Casting with High Strength–Ductility Synergy Properties: A Review"

_materials, 2023, doi:10.3390/ma16051954_

Round 1

Reviewer 1 Report

The review of the microstructures and strength-ductility synergy properties of high-pressure die casting magnesium alloys is written in detail, with many supporting references, adequate organization of the sub-chapters, and relevant characterization of the alloys. However, there are some issues that should be fixed before the review is published. Specifically:

1. English is very poor, with many grammatical and style errors (few examples of many are the sentence in lines 295-299, 615-618, 639-642). Sometimes, it even affects the legibility of the sentences (for example, the sentence in Abstract, lines 22-24). Please, review and improve English considerably.

2. It would be beneficial for the wider audience interested in topic if the short description of the measurement/analysis methods and their purposes are given.

3. There are too many overlapping points and symbols in Figure 1. It should be separated into several figures. Furthermore, references should be placed next to the text describing and discussing Figure 1, not in the figure caption.

4. Figures should be placed as near as possible to the text describing them.

5. Some references are old. If possible, include more recent references.

Author Response

There are too many figures, too detailed with specific microstructures to a specific case. Reduce the number of figures, and give more summary tables instead.

A: Thank you for this relevant comment. In our revised manuscript, only 17 figures are remained and a summary table was given as:

Table 1 Grain size and dominant intermetallic phases in various HPDC Mg alloys.

Alloy

Grain size (mm)

Dominant intermetallic phases

Reference

AE44

~14

Acicular/needlelike Al11RE3, blocky Al2RE

44,45

ALa44

7-16

Acicular/needlelike Al11La3

83,84

ALaM440

~15.6

Acicular h-Al3La (or (Mg,Al)3La)

66,119

ACe44

~14

Acicular/needlelike Al11Ce3, blocky Al2Ce

96

APr44

~8.5

Quadrangle/polygon Al2Pr, acicular/petal-like Al11Pr3

97

ANd44

~10.5

Quadrangle Al2Nd, dendritic/ coarse-surface Al11Nd3

85,95

ASm44

~9.6

Petaloid Al2Sm

86

ASm46

~6.3

Petaloid/blocky/short-rod-like Al2Sm, dendritic Al11Sm3

87

AGd44

~5.9

Petaloid/blocky Al2Gd

88

AE432

~6.1

Acicular/flaky h-Al3RE, blocky/spindle-shapedAl2RE, claviform Al7RE3

69,89

AEX series

~7

Lamellae (Mg,Al)2Ca, lath-like Al11RE3, rod-like h-Al3RE, blocky Al2.12RE0.88

68,81,108

AEJ series

~13.6

blocky Al4Sr or Mg8Al4Sr, lamellae Mg17Sr2

80,108-110

ALa44+B

10.7

Acicular Al11RE3, blocky Al2RE

82

Mg-RE series

~5.1

Net-work Mg12RE, lamellae Mg3RE, plate-like b-series precipitates

91,140

ZLaW423

~6.1

Aggregated lamellae Mg12RE and blocky W phase, lath-like LPSO

93

Mg-RE-Al

a1=10-30, a2=2-10,

Lamellae Mg12RE, blocky Al2RE3, precipitated AlMnRE cluster/Al2RE

71,94

AX/J series

~6.1

Lamellae (Mg,Al)2Ca/Mg17Sr2, net-work Mg2Ca/Al4Sr, precipitated Al2Ca

90,92,156-160

ZA88

a1=~5.6, a2=~0.67,

Nano-sized icosahedral quasi-crystal phase

65

Review paper should not just contain published figures, it is expected the authors should create ~3 new figures of their own, based on the published data. And more importantly, in a general and summary form, not in the very specific form of a microstructures.

A: Thank you for this relevant comment. In our revised manuscript, all figures were re-created or modified.

What is the novelty of this paper compared with other relevant review papers published?

A: Thank you for this comment. This paper is dealing with the relationship between microstructural characteristics, mainly intermetallic phases, and the strength-ductility synergy. As stated in our revised manuscript, this paper reviews the strength and the ductility of various HPDC Mg alloys at room temperature and states the corresponding microstructural characteristics, aimed to serve a guide for the future development of HPDC Mg alloys with high-strength and high-ductility.

As mentioned above, there is a lack of summarised table for reference of the readers.

A: Thank you for this relevant comment. We added a summarized table in our revised manuscript as presented in the above.

What is the future development trend?

A: Thank you for this relevant comment. In the 7 section, we give the characteristics and the points of further development for AE system and Mg-RE system, separately. In our revised manuscript, we further modified the future development trend as (page 19 line 32-45):

Modifying alloy’s chemical compositions is the key for designing new HPDC alloys with high-strength and high-ductility, and the following heat treatments at relatively low temperatures are efficient for enhancing alloys’ properties particularly yield strength. However, aimed to promote the wide applications of HPDC Mg alloys in such as automobile and aerospace industries, other properties such as creep resistance, thermal conductivity, corrosion resistance, castability should also be appreciated during future development. The combination property is much more important. The RE containing Mg alloys might be the most potential candidate for the future wide applications. Generally, complex intermetallic phases on grain boundaries are expected to accomplish moderate strength and high ductility, and continuous or semi-continuous lamellae intermetallic skeleton is for ultra-high strength. As mentioned in the above, micro-alloying and multi-alloying exhibit excellent effect on modifying microstructures of the RE-containing Mg alloys and should be paid attention to in the future investigations.Microstructural characters should be Microstructural characteristics.

A: Thank you for this pertinent comment. In our revised manuscript, we revised all “microstructural characters” as “microstructural characteristics”.

In addition to 2-6, add more refs about the industrial applications of mg, 10.1016/j.jma.2023.01.002; 10.1016/j.jma.2022.11.012.

A: Thank you for this relevant comment. In our revised manuscript, we have added some relevant references about the industrial applications of Mg, including these two papers.

Reviewer 2 Report

The manuscript requires few modification:

1.      There are many group citations [2-6, 11-14, 16-24, 28-40, 41-51 and so on]. It is recommended to report on individual contributions.

2.      There are many influencing parameters (melt and die temperature, different pressure modes and so on) in HPDC and those effects on microstructure must be reported briefly.

3.      Optimization of parameters on microstructure is not reported.

4.      Microstructure changes with grain refiners and modifiers is interesting.  

5.      Statistical relationship between structure and property need to be established.

Author Response

  1. There are many group citations [2-6, 11-14, 16-24, 28-40, 41-51 and so on]. It is recommended to report on individual contributions.

A: Thank you for this comment. In our revised manuscript, we have tried our best to report the references individually, but in many cases one viewpoint were supported by several references.

  1. There are many influencing parameters (melt and die temperature, different pressure modes and so on) in HPDC and those effects on microstructure must be reported briefly.

A: Thank you for this relevant comment. In our revised manuscript, we have revised it as (page 2 line 10-18):

On the other hand, defects such as gas pores and shrinkage would unavoidably form in castings [16]. Hence, huge work has been carried out in order to reduce or even eliminate these die casting defects by controlling melt and die temperature [17], optimizing HPDC parameters [18,19], or using vacuum-assisted system [20]. Generally, defects would seriously deteriorate the mechanical properties of HPDC Mg alloys, and the property difference reported by different authors might be from these defects. However, considering the main topic on intermetallic phases, optimization of HPDC parameters on microstructures and mechanical properties was not discussed in this paper.

The added references:

[16] Yu, W.; Cao, Y.; Li, X.; Guo, Z.; Xiong, S. Determination of interfacial heat transfer behavior at the metal/shot sleeve of high pressure die casting process of AZ91D alloy. J. Mater. Sci. Technol. 2017, 33, 52–58.

[17] Xiong, S.; Li, X.; Guo, Z. Influence of melt flow on the formation of defect band in high pressure die casting of AZ91D magnesium alloy. Mater. Charact. 2017, 129, 344–352.

[18] Biswas, S.; Sket, F. Relationship between the 3D porosity and β-phase distributions and the mechanical properties of a high pressure die cast AZ91 Mg alloy. Metall. Mater. Trans. A 2013, 44, 4391–4403.

[19] Zhang, T.; Yu, W.; Ma, C.; Chen, W.; Zhang, L.; Xiong, S. The effect of different high pressure die casting parameters on 3D microstructure and mechanical properties of AE44 magnesium alloy, J. Magnes. Alloys, doi: 10.1016/j.jma.2022.05.001.

[20] Hou, Y.; Wu, M.; Tian, B.; Li, X.; Xiong, S. Characteristics and formation mechanisms of defect bands in vacuum-assisted high-pressure die casting AE44 alloy. Trans. Nonfer. Metals Sco. China 2022, 32(6) 1852-1865.

  1. Optimization of parameters on microstructure is not reported.

A: Thank you for this relevant comment. As mention in the above, we modified the corresponding description and stated that “However, considering the main topic on intermetallic phases, optimization of HPDC parameters on microstructures and mechanical properties was not discussed in this paper.”.

  1. Microstructure changes with grain refiners and modifiers is interesting.

A: Thank you for this relevant comment. In our revised manuscript, grain size was listed in Table 1 and discussed in the 7 section as “(page 39 line 16-44) Grain refinement can simultaneously improve alloys’ strength and ductility at room temperatures. However, the grain size of HPDC Mg alloys are comparable in a range of 5-20 mm (except thin-wall components), and no clear relationship between fine-grain and high strength-ductility. Inversely, the HPDC Mg alloys with high-strength exhibit different intermetallic characteristics with those in the alloys with high-ductility.”.

Table 1 Grain size and dominant intermetallic phases in various HPDC Mg alloys.

Alloy

Grain size (mm)

Dominant intermetallic phases

Reference

AE44

~14

Acicular/needlelike Al11RE3, blocky Al2RE

44,45

ALa44

7-16

Acicular/needlelike Al11La3

83,84

ALaM440

~15.6

Acicular h-Al3La (or (Mg,Al)3La)

66,119

ACe44

~14

Acicular/needlelike Al11Ce3, blocky Al2Ce

96

APr44

~8.5

Quadrangle/polygon Al2Pr, acicular/petal-like Al11Pr3

97

ANd44

~10.5

Quadrangle Al2Nd, dendritic/ coarse-surface Al11Nd3

85,95

ASm44

~9.6

Petaloid Al2Sm

86

ASm46

~6.3

Petaloid/blocky/short-rod-like Al2Sm, dendritic Al11Sm3

87

AGd44

~5.9

Petaloid/blocky Al2Gd

88

AE432

~6.1

Acicular/flaky h-Al3RE, blocky/spindle-shapedAl2RE, claviform Al7RE3

69,89

AEX series

~7

Lamellae (Mg,Al)2Ca, lath-like Al11RE3, rod-like h-Al3RE, blocky Al2.12RE0.88

68,81,108

AEJ series

~13.6

blocky Al4Sr or Mg8Al4Sr, lamellae Mg17Sr2

80,108-110

ALa44+B

10.7

Acicular Al11RE3, blocky Al2RE

82

Mg-RE series

~5.1

Net-work Mg12RE, lamellae Mg3RE, plate-like b-series precipitates

91,140

ZLaW423

~6.1

Aggregated lamellae Mg12RE and blocky W phase, lath-like LPSO

93

Mg-RE-Al

a1=10-30, a2=2-10,

Lamellae Mg12RE, blocky Al2RE3, precipitated AlMnRE cluster/Al2RE

71,94

AX/J series

~6.1

Lamellae (Mg,Al)2Ca/Mg17Sr2, net-work Mg2Ca/Al4Sr, precipitated Al2Ca

90,92,156-160

ZA88

a1=~5.6, a2=~0.67,

Nano-sized icosahedral quasi-crystal phase

65

  1. Statistical relationship between structure and property need to be established.

A: Thank you for this relevant comment. In our revised manuscript, we revised it as “(page 19 line 30-36) The RE containing Mg alloys might be the most potential candidate for the future wide applications. Generally, complex intermetallic phases on grain boundaries are expected to accomplish moderate strength and high ductility, and continuous or semi-continuous lamellae intermetallic skeleton is for ultra-high strength. As mentioned in the above, micro-alloying and multi-alloying exhibit excellent effect on modifying microstructures of the RE-containing Mg alloys and should be paid attention to in the future investigations.”.

Reviewer 3 Report

Reviewer’s Comments:

The manuscript “Microstructures and strength-ductility synergy properties of high-pressure die casting magnesium alloys: a review” is a very interesting work. In this work, In response to the increased emphasis on reducing carbon emissions, the demand for lightweight, high-performance structural materials is quickly increasing, and magnesium alloys, because of their lowest density among the common engineering metals, have demonstrated considerable advantages and prospective applications in contemporary industry. High-pressure die casting (HPDC) owning high efficiency and low production cost is the most extensively utilized technique in commercial magnesium alloy applications. High room-temperature strength-ductility of HPDC magnesium alloys plays an important role in safe using in particularly automotive and aerospace industries. With respect to HPDC magnesium alloys, mechanical properties are highly relied on microstructural characters, particularly intermetallic phases, which is further dependent on alloys’ chemical compositions. Therefore, further alloying on traditional HPDC magnesium alloys such as Mg-Al, Mg-RE and Mg-Zn-Al systems, is the most adopted method to further improve their mechanical properties. Different alloying elements lead to different intermetallic phases or with different morphologies or with different crystal structures, of which both would generate helpful or harmful effects on alloys strength or ductility. While I believe this topic is of great interest to our readers, I think it needs major revision before it is ready for publication. So, I recommend this manuscript for publication with major revisions.

1. In this manuscript, the authors did not explain the importance of the strength-ductility the introduction part. The authors should explain the importance of strength-ductility.

2) Title: The title of the manuscript is not impressive. It should be modified or rewritten it.

3) Correct the following statement “This paper gives a comprehensive review of the strength-ductility synergy property and the corresponding intermetallic phases including crystal structures, morphologies, and faults, of various HPDC magnesium alloys with good properties, and provided insight into designing high-performance HPDC magnesium alloys”.

4) Keywords: The strength-ductility is missing in the keywords. So, modify the keywords.

5) Introduction part is not impressive. The references cited are very old. So, Improve it with some latest literature like 10.1016/j.eurpolymj.2021.110783, 10.1016/j.chemosphere.2022.133772.

6) The authors should explain the following statement with recent references, “As the Pr addition increases, more large blocky particles appeared in matrix, and some new particles such as with petal-like morphology generated”.

7) Add space between magnitude and unit. For example, in synthesis “21.96g” should be 21.96 g. Make the corrections throughout the manuscript regarding values and units.

8) The author should provide reason about this statement “The intermetallic phases become diversifed, but few new intermetallic particles were introduces except the hammer-like Al2RE particles”.

9. Comparison of the present results with other similar findings in the literature should be discussed in more detail. This is necessary in order to place this work together with other work in the field and to give more credibility to the present results.

10) Conclusion part is very long. Make it brief and improve by adding the results of your studies.

11) There are many grammatic mistakes. Improve the English grammar of the manuscript.

Author Response

The manuscript “Microstructures and strength-ductility synergy properties of high-pressure die casting magnesium alloys: a review” is a very interesting work. In this work, in response to the increased emphasis on reducing carbon emissions, the demand for lightweight, high-performance structural materials is quickly increasing, and magnesium alloys, because of their lowest density among the common engineering metals, have demonstrated considerable advantages and prospective applications in contemporary industry. High-pressure die casting (HPDC) owning high efficiency and low production cost is the most extensively utilized technique in commercial magnesium alloy applications. High room-temperature strength-ductility of HPDC magnesium alloys plays an important role in safe using in particularly automotive and aerospace industries. With respect to HPDC magnesium alloys, mechanical properties are highly relied on microstructural characters, particularly intermetallic phases, which is further dependent on alloys’ chemical compositions. Therefore, further alloying on traditional HPDC magnesium alloys such as Mg-Al, Mg-RE and Mg-Zn-Al systems, is the most adopted method to further improve their mechanical properties. Different alloying elements lead to different intermetallic phases or with different morphologies or with different crystal structures, of which both would generate helpful or harmful effects on alloys strength or ductility. While I believe this topic is of great interest to our readers, I think it needs major revision before it is ready for publication. So, I recommend this manuscript for publication with major revisions.

  1. In this manuscript, the authors did not explain the importance of the strength-ductility the introduction part. The authors should explain the importance of strength-ductility.

A: Thank you for this relevant comment. In our revised manuscript, we revised it as “(page 14 line 24-30) In engineering applications, the strength-ductility synergy of the structural materials plays an important role in safe using [55-57]. However, the low strength and poor ductility of Mg alloys seriously limited the wide applications particularly in power-train components [57].”

The added references:

[55] Zhou, W.; Lin, J.; Dean, T.A. Microstructure and mechanical properties of curved AZ31 magnesium alloy profiles produced by differential velocity sideways extrusion. J. Magnes. Alloys, doi: 10.1016/j.jma.2022.11.012.

[56] Fan, M.; Zhang, Z.; Cui, Y.; Liu, L.; Liu, Y.; Liaw, P.K. Achieving strength and ductility synergy via a nanoscale superlattice precipitate in a cast Mg-Y-Zn-Er alloy. Int. J. Plast. 2023, 163, 103558.

[57] Feng, J.; Zhang, L.; Zhang, Y.; Feng, G.; Wang, C.; Fang, W. Improvement of strength and ductility synergy and the elimination of the yield point phenomenon of ultrafine-grained Mg-Al-Zn-Ag alloy sheet. J. Mater. Research Technol. 2023, 23, 2900-2910.

2) Title: The title of the manuscript is not impressive. It should be modified or rewritten it.

A: Thank you for this relevant comment. In our revised manuscript, we revised the title as “Microstructural characteristics of high-pressure die casting with high strength-ductility synergy properties: a review”.

3) Correct the following statement “This paper gives a comprehensive review of the strength-ductility synergy property and the corresponding intermetallic phases including crystal structures, morphologies, and faults, of various HPDC magnesium alloys with good properties, and provided insight into designing high-performance HPDC magnesium alloys”.

A: Thank you for this relevant comment. In our revised manuscript, we revised it as “This paper focuses on microstructural characteristics, mainly intermetallic phases (components and morphologies), of various HPDC Mg alloys with good strength-ductility synergy, aimed to provide insight into designing high-performance HPDC Mg alloys.”.

4) Keywords: The strength-ductility is missing in the keywords. So, modify the keywords.

A: Thank you for this relevant comment. In our revised manuscript, we revised the keywords as “Mg alloys; high-pressure die casting; strength; ductility; intermetallic phase”.

5) Introduction part is not impressive. The references cited are very old. So, Improve it with some latest literature like 10.1016/j.eurpolymj.2021.110783, 10.1016/j.chemosphere.2022.133772.

A: Thank you for this relevant comment. In our revised manuscript, we have added some latest references including these two papers.

6) The authors should explain the following statement with recent references, “As the Pr addition increases, more large blocky particles appeared in matrix, and some new particles such as with petal-like morphology generated”.

A: Thank you for this pertinent comment. In our revised manuscript, we have revised it as “As the Pr addition increases, more large blocky particles appeared in matrix, and some new particles such as with petal-like morphology generated [97].”.

[97] Zhang, J.; Liu, K.; Fang, D.; Qiu, X.; Yu, P.; Tang, D.; Meng, J. Microstructures, mechanical properties and corrosion behavior of high-pressure die-cast Mg–4Al–0.4Mn–xPr (x = 1, 2, 4, 6) alloys. J. Alloys. Compd. 2009, 480, 810-819.

7) Add space between magnitude and unit. For example, in synthesis “21.96g” should be 21.96 g. Make the corrections throughout the manuscript regarding values and units.

A: Thank you for this pertinent comment. In our revised manuscript, we have added space between magnitude and unit.

8) The author should provide reason about this statement “The intermetallic phases become diversifed, but few new intermetallic particles were introduces except the hammer-like Al2RE particles”.

A: Thank you for this pertinent comment. In our revised manuscript, we have revised it as “The intermetallic phases become diversified with many particles with new morphologies being appeared, but only the hammer-like Al2RE particles were not observed in traditional Mg-Al-RE based alloys.”.

  1. Comparison of the present results with other similar findings in the literature should be discussed in more detail. This is necessary in order to place this work together with other work in the field and to give more credibility to the present results.

A: Thank you for this comment. We carefully described the novelty and credibility of this review paper in our revised manuscript.

10) Conclusion part is very long. Make it brief and improve by adding the results of your studies.

A: Thank you for this comment. Our manuscript is a review paper and the conclusion part includes summary and outlook.

11) There are many grammatic mistakes. Improve the English grammar of the manuscript.

A: Thank you for this relevant comment. We have carefully checked the English grammar throughout our revised manuscript.

Reviewer 4 Report

There are too many figures, too detailed with specific microstructures to a specific case. Reduce the number of figures, and give more summary tables instead.

Review paper should not just contain published figures, it is expected the authors should create ~3 new figures of their own, based on the published data. And more importantly, in a general and summary form, not in the very specific form of a microstructures.

What is the novelty of this paper compared with other relevant review papers published?

As mentioned above, there is a lack of summarised table for reference of the readers.

What is the future development trend?

Microstructural characters should be Microstructural characteristics

In addition to 2-6, add more refs about the industrial applications of mg, 10.1016/j.jma.2023.01.002; 10.1016/j.jma.2022.11.012

Author Response

(The authors gave the same response as above.)

Round 2

Reviewer 2 Report

Congratulations for your excellent review and revision.

Reviewer 4 Report

Thanks for the detailed response, I have no further comments